# Enhanced Cardiac CaMKII Oxidation and CaMKII-Dependent SR Ca Leak in Patients with Sleep-Disordered Breathing

**DOI:** 10.3390/antiox11020331

**Published:** 2022-02-08

**Authors:** Michael Arzt, Marzena A. Drzymalski, Sarah Ripfel, Sebastian Meindl, Alexander Biedermann, Melanie Durczok, Karoline Keller, Julian Mustroph, Sylvia Katz, Maria Tafelmeier, Simon Lebek, Bernhard Flörchinger, Daniele Camboni, Sigrid Wittmann, Johannes Backs, Christof Schmid, Lars S. Maier, Stefan Wagner

**Affiliations:** 1Department of Internal Medicine II, University Hospital Regensburg, 93053 Regensburg, Germany; michael.arzt@ukr.de (M.A.); Marzena.Drzymalski@gmail.com (M.A.D.); sarah.ripfel@klinik.uni-regensburg.de (S.R.); tt_basti@yahoo.de (S.M.); alexanderbiedermann@gmx.net (A.B.); melanie.durczok@ukr.de (M.D.); karoline.keller92@outlook.de (K.K.); julian.mustroph@ukr.de (J.M.); maria.tafelmeier@klinik.uni-regensburg.de (M.T.); simon.lebek@ukr.de (S.L.); lars.maier@ukr.de (L.S.M.); 2Department of Molecular Cardiology and Epigenetics, University of Heidelberg, 69120 Heidelberg, Germany; Sylvia.Katz@med.uni-heidelberg.de (S.K.); johannes.backs@med.uni-heidelberg.de (J.B.); 3German Centre for Cardiovascular Research (DZHK), Partner Site Heidelberg/Mannheim, 69120 Heidelberg, Germany; 4Department of Cardiothoracic Surgery, University Hospital Regensburg, 93053 Regensburg, Germany; Bernhard.Floerchinger@ukr.de (B.F.); daniele.camboni@klinik.uni-regensburg.de (D.C.); Christof.Schmid@ukr.de (C.S.); 5Department of Anesthesiology, University Hospital Regensburg, 93053 Regensburg, Germany; sigrid.wittmann@ukr.de

**Keywords:** calcium–calmodulin-dependent protein kinase type II, sleep apnea, atrial fibrillation, sarcoplasmic reticulum Ca leak

## Abstract

Background: Sleep-disordered breathing (SDB) is associated with increased oxidant generation. Oxidized Ca/calmodulin kinase II (CaMKII) can contribute to atrial arrhythmias by the stimulation of sarcoplasmic reticulum Ca release events, i.e., Ca sparks. Methods: We prospectively enrolled 39 patients undergoing cardiac surgery to screen for SDB and collected right atrial appendage biopsies. Results: SDB was diagnosed in 14 patients (36%). SDB patients had significantly increased levels of oxidized and activated CaMKII (assessed by Western blotting/specific pulldown). Moreover, SDB patients showed a significant increase in Ca spark frequency (CaSpF measured by confocal microscopy) compared with control subjects. CaSpF was 3.58 ± 0.75 (SDB) vs. 2.49 ± 0.84 (no SDB) 1/100 µm^−1^s^−1^ (*p* < 0.05). In linear multivariable regression models, SDB severity was independently associated with increased CaSpF (B [95%CI]: 0.05 [0.03; 0.07], *p* < 0.001) after adjusting for important comorbidities. Interestingly, 30 min exposure to the CaMKII inhibitor autocamtide-2 related autoinhibitory peptide normalized the increased CaSpF and eliminated the association between SDB and CaSpF (B [95%CI]: 0.01 [−0.1; 0.03], *p* = 0.387). Conclusions: Patients with SDB have increased CaMKII oxidation/activation and increased CaMKII-dependent CaSpF in the atrial myocardium, independent of major clinical confounders, which may be a novel target for treatment of atrial arrhythmias in SDB.

## 1. Introduction

Atrial fibrillation (AF) is the most common cardiac arrhythmia, affecting approximately 4.5 million people in the European Union and 2.3 million people in North America [1,2,3]. The incidence of AF is expected to increase dramatically by 2050, as the general population ages and becomes more obese [4]. AF is associated with substantially increased morbidity and mortality, owing to the concomitant risk of embolic stroke and worsening heart failure (HF) [1]. At the level of the atrial cardiomyocyte, electrical, structural and Ca-handling remodeling have been suggested to contribute to the induction and perpetuation of AF [5,6]. Previous evidence suggests that spontaneous sarcoplasmic reticulum (SR) Ca release events (i.e., Ca sparks) are a major trigger for AF [7,8,9]. In particular, increased activity of Ca/calmodulin kinase II (CaMKII) and consequent hyperphosphorylation of the SR Ca release channel RyR2 has been identified in the right atrial myocardium of patients with AF [8,9].

Sleep-disordered breathing (SDB) is a widespread disease of high socio-economic importance owing to its clinical corollaries [6]. Important consequences of SDB are atrial arrhythmias including AF [10], stroke [11,12] and hypertension [13]. SDB has also been associated with recurrence of AF after electrical cardioversion [14] or ablation of AF [15]. The method of choice for treatment of SDB is continuous positive airway (CPAP) therapy, but there is increasing evidence that CPAP therapy does not reduce the burden of AF [16], as previously thought. Unfortunately, the pathophysiological mechanisms of arrhythmogenesis in SDB are poorly understood. Recently we have shown that patients with SDB have increased amounts of reactive oxygen species (ROS) in the atrial myocardium [17]. Oxidative stress is induced in SDB by intermittent hypoxemia with cyclical changes of hypoxemia with reoxygenation [18,19]. Interestingly, oxidation of CaMKII is an important mechanism of CaMKII activation [20], and oxidized CaMKII has been shown to contribute to AF in a murine model [21]. Moreover, recently we have shown an enhanced activity of CaMKII that contributes to pro-arrhythmic activity in atrial myocardium of cardiovascular patients with SDB [17].

Therefore, we tested the hypothesis that patients with SDB have more oxidized CaMKII in the atrium, and that CaMKII-dependent Ca sparks are more frequent in patients with SDB. This has therapeutic relevance, since CaMKII inhibitors are already in preclinical development [22].

## 2. Materials and Methods

### 2.1. Study Design

Patients undergoing elective coronary artery bypass graft (CABG) surgery were prospectively included for tissue donation (Figure 1). Exclusion criteria were known sleep apnea and CPAP therapy. Functional measurements of atrial cardiomyocytes and protein analysis of atrial biopsies were performed by blinded investigators.

### 2.2. Study Approval

This research was approved by the ethics committee and performed in accordance with the Declaration of Helsinki (first released in 1964, most recent revision 2013). Patients provided written consent prior to tissue donation.

### 2.3. Assessment of SDB by Polygraphy

Nasal flow, pulse oximetry and thoracal breathing effort were measured using the ApneaLink device (ResMed, Sydney, Australia) that has been validated in several studies for screening of SDB [23,24,25,26,27,28]. Comparing ApneaLink (using automatic scoring) to the gold standard polysomnography (PSG) in patients without known heart disease, studies have reported a sensitivity of 73–94% and a specificity of 85–95% using an AHI cut-off value of 15/h [29]. As described previously [29,30], apnea was defined as a ≥80% decrease in airflow for ≥10 s; hypopnea was defined as a decrease in airflow by ≥50–80% versus baseline for ≥10 s; desaturation was defined as a ≥4% decrease in oxygen saturation; the oxygen desaturation index (4%) is the number of desaturations per hour. SDB was defined as an apnea-hypopnea index (AHI) ≥15 per hour.

### 2.4. Confocal Ca Measurements

Ca sparks were assessed by confocal microscopy using the Ca dye fluo-4 acetoxymethylester. Isolated myocytes were loaded with 10 µmol/L fluo-4 acetoxymethylester (Molecular Probes) for 12 min at room temperature and mounted on an inverted laser scanning confocal microscope (Zeiss LSM 7) for measurement.

### 2.5. Statistical Analysis

Continuous data are expressed as mean ± standard deviation (SD). Please refer to the Appendix A for detailed methods.

## 3. Results

### 3.1. Patients

Thirty-nine patients with coronary heart disease undergoing elective CABG surgery fulfilled the inclusion and exclusion criteria for this analysis (Figure 1).

Moderate to severe SDB (AHI ≥ 15/h) was diagnosed in 36% of patients. Comparing the baseline characteristics of both groups, patients with SDB had a significantly lower left ventricular ejection fraction (LVEF, *p* = 0.004) and a higher rate of left atrial dilation (*p* = 0.011, Table 1). Interestingly, SDB patients showed a significantly greater overall prevalence of AF (paroxysmal and permanent combined, Table 1). AF was present in 43% of SDB patients vs. 12% in the group without SDB (*p* = 0.028, Table 1). A history of stroke was significantly more common in SDB patients compared to those without SDB (*p* = 0.028, Table 1). The remaining variables were not significantly different between cohorts. However, age, the need for additional surgical valve replacement and NT-proBNP levels were numerically higher in the SDB group. In contrast, ACE inhibitor/angiotensin receptor blocker (ACEi/ARB) therapy was less frequent in the SDB group (Table 1). Gender representation in the SDB group was exclusively male.

By definition, patients in the SDB group had moderate to severe SDB, with a significantly higher oxygen desaturation index (4%) (*p* < 0.001), indicating intermittent hypoxemia (Table 2). In the SDB group, the majority of apneas was obstructive in nature, and Cheyne–Stokes respiration was present in only 9% of total recording time (Table 2).

### 3.2. Increased CaMKII Oxidation and Activity in Patients with SDB

Intermittent hypoxemia results in ROS generation. To investigate if CaMKII is more oxidized in SDB patients, which would lead to increased CaMKII activity, we measured the level of oxidized CaMKII in homogenates from right atrium (RA) biopsies (Figure 2A,B). Compared to patients without SDB, ox-CaMKII levels (relative to total CaMKII expression) were significantly increased in biopsies from patients with SDB (Figure 2A,B). The ox-CaMKII/CaMKII ratio doubled from 0.74 ± 0.36 (no SDB, n = 15) to 1.51 ± 0.76 (SDB, n = 8, *p* < 0.05). Interestingly, there was a significant positive correlation between the severity of SDB (AHI) and CaMKII oxidation (B [95%CI]: 0.043 [0.021; 0.065], n = 23, R^2^ = 0.447, *p* < 0.05, Figure 2C, Appendix A in Appendix A). Moreover, CaMKII oxidation also significantly correlated with oxygen desaturation index (ODI), another measure of intermittent hypoxia. In contrast, parameters of sustained hypoxia such as minimal nocturnal oxygen saturation and percentage of time below 90% saturation were not found more frequently in SDB patients (Table 2) and were not associated with the extent of CaMKII oxidation (Appendix A in Appendix A).

In accordance, the activity of CaMKII (measured as described previously [31]) was significantly enhanced in RA biopsies of patients with SDB (Figure 2D). CaMKII activity increased from 1.34 ± 0.74 (no SDB, n = 13) to 2.28 ± 0.67 (SDB, n = 6, *p* < 0.05). There was also a significant correlation between CaMKII activity and AHI (linear regression analysis, B [95%CI]: 0.066 [0.023; 0.109], n = 19, R^2^ = 0.38, *p* < 0.05, Figure 2E).

### 3.3. Ca Spark Frequency Is Increased in Patients with SDB

To investigate whether the SDB-dependent increase in CaMKII activity may be accompanied by disturbed intracellular Ca handling, Ca sparks were measured by confocal microscopy in atrial cardiomyocytes isolated as previously described [9,32]. Figure 3 shows confocal line scans of cardiomyocytes from two representative individual patients from both groups. The frequency of diastolic Ca release events, i.e., Ca spark frequency (CaSpF), was significantly increased in patients with SDB (Figure 3B). CaSpF (in 1/100 µm^−1^s^−1^) increased from 2.49 ± 0.84 (no SDB, n = 25) to 3.58 ± 0.75 (SDB, n = 14, *p* < 0.05, Figure 3B).

Moreover, CaSpF correlated significantly with the severity of SDB (AHI), as assessed by linear correlation (R^2^ = 0.44, B [95%CI]: 0.052 [0.032; 0.071], n = 39, *p* < 0.001; Table 3, and Appendix A in Appendix A). In accordance with CaMKII oxidation, CaSpF also significantly correlated with ODI, but not with minimal nocturnal oxygen saturation nor percentage of time below 90% saturation (Table 3). We have shown previously that stimulation of the angiotensin 1 receptor results in increased CaSpF in a CaMKII-dependent fashion [33]. Interestingly, univariate linear regression analysis indicate that ACEi/ARB therapy was indeed significantly associated with a decreased CaSpF (Table 3) but not with decreased CaMKII oxidation (Appendix A in Appendix A).

### 3.4. CaMKII Inhibition Reduces SDB-Dependent Ca Spark Frequency

To test if enhanced SDB-dependent CaMKII activity is responsible for the increase in CaSpF, we also exposed isolated atrial cardiomyocytes to the selective CaMKII inhibitor AIP (2 µmol/L, 30 min). In the presence of AIP, CaSpF was significantly reduced in both groups (Figure 3B, spaghetti plots), masking the difference between patients with and without SDB. Similarly, linear regression analysis revealed a loss of correlation of AHI and CaSpF in the presence of AIP (R^2^ = 0.04, *p* = 0.355; B[CI]: 0.01 [−0.01; 0.03], *p* = 0.355). In accordance, AIP blocked the SDB-dependent enhancement of diastolic SR Ca leak (Appendix A). Interestingly, AIP significantly increased the amplitude of individual Ca sparks in SDB patients, indicating that SR Ca content may increase with CaMKII inhibition (Appendix A).

### 3.5. SR Ca Content Was Reduced in Patients with SDB

To find out if enhanced SR Ca leak in SDB patients could be linked to reduced SR Ca content, which would impair atrial contractility, we rapidly exposed atrial cardiomyocytes to caffeine. Figure 4A illustrates representative line scan images of caffeine-induced Ca transients (ΔF/F_0_) from two representative individual patients. The caffeine-transient amplitude was significantly reduced in atrial myocytes of patients with SDB (Figure 4B). ΔF/F_0_ was reduced from 8.03 ± 2.39 (no SDB, n = 14) to 4.23 ± 1.60 (SDB, n = 7, *p* < 0.05, Figure 4B). In addition, there was a significant negative correlation between AHI and caffeine-transient amplitude (R^2^ = 0.307, B [95%CI]: −1.04 [−1.79; −0.29], n = 21, *p* < 0.05, Table 4, Appendix A). Conversely, acute inhibition of CaMKII with AIP tended to increase the caffeine-transient amplitude in SDB patients (Figure 4B). Consistently, linear regression analysis revealed a loss of correlation of AHI and CaSpF in the presence of AIP (R^2^ = 0.001; *p* = 0.896; B (CI); 0.02 (−0.29; 0.26), n = 17, *p* = 0.896).

### 3.6. SDB-Dependent Atrial Remodeling Was Also Observed in Patients with Sinus Rhythm

To exclude the possibility that pre-existing AF (irrespective of cause) and not the severity of SDB underlies the observed increase in CaMKII activity and changes in intracellular Ca handling, we repeated the analysis after omitting all patients with either pre-existing paroxysmal or persistent AF (Appendix A). Notably, after exclusion of patients with AF, the SDB-dependent enhancement of CaMKII oxidation and activity was completely maintained (Appendix A). Moreover, the SDB-dependent increase in CaSpF and the reduction in caffeine-transient amplitude were also not affected by the removal of patients with AF from the analysis set (Appendix A).

To further investigate potential clinical confounders, we performed multivariate linear regression with CaMKII oxidation (Appendix A in Appendix A), CaSpF (Table 3) and caffeine-induced Ca transients (Table 4). Model I accounts for age, gender and BMI, and model II additionally accounts for NT-pro BNP, diabetes, AF and ACEi/ARB therapy.

In these models, AHI remained the only significant predictor of CaMKII oxidation, CaSpF and caffeine-induced Ca transients. Similarly to AHI, the obstructive apnea index (OAI), the central apnea index (CAI) and the frequency of Cheyne–Stokes respiration were associated with increased CaSpF, independent of known potential confounders (model I, OAI: (B [95%CI]): 0.118 [0.028; 0.209], *p* = 0.012, Appendix A; CAI: (B [95%CI]): 0.06 [0.023; 0.096], *p* = 0.002, Appendix A, Cheyne–Stokes respiration: (B [95%CI]): 0.291 [−0.180; 0.600], *p* = 0.064, Appendix A).

## 4. Discussion

The major novel finding of this study is the observation that patients with SDB have increased atrial CaMKII-dependent SR-Ca leak and decreased SR-Ca content, which can be reversed by acute CaMKII inhibition. This SDB-dependent CaMKII-related SR Ca leak correlates with the severity of SDB and is independent of important confounding clinical factors such as preexisting AF, HF or diabetes.

### 4.1. SDB Results in Increased CaMKII Oxidation

Increased activity of CaMKII has been causally linked to contractile dysfunction and arrhythmias in patients with AF [8,9] and HF [34,35,36,37]. However, the mechanisms of activation remain insufficiently understood. In addition to Ca–calmodulin-dependent stimulation, oxidation of CaMKII is an important alternative CaMKII activation pathway [20], which is critical for the regulation of intracellular Na and Ca homeostasis [38,39,40]. The relevance of oxidized CaMKII for atrial arrhythmias has been confirmed in a murine model of AF [21]. Interestingly, SDB is characterized by intermittent arterial hypoxemia with cyclical changes of hypoxemia with reoxygenation. As a result, the production of ROS is increased [18,19]. Recently we have shown that patients with SDB have increased amounts of reactive oxygen species (ROS) in atrial myocardium [17], which was in accordance murine SDB models (chronic intermittent hypoxia), but also patients [17,19,41,42]. Here we report increased CaMKII oxidation and CaMKII activation in atrial myocardium of patients with SDB. Moreover, the magnitude of CaMKII oxidation correlates significantly with the severity of the SDB (measured by AHI), which suggests that CaMKII oxidation may be a major CaMKII activation pathway in SDB. Thus, inhibition of CaMKII oxidation may represent an even more specific target for the treatment or prevention of arrhythmias in patients with SDB. This is particularly important since ATP-competitive CaMKII inhibitors with high CaMKII selectivity and good bioavailability, but nonspecific inhibition of CaMKII activity regardless of their activation pathway, are currently being developed [22].

### 4.2. SDB Patients Show Increased CaMKII-Dependent SR Ca Leak and Reduced SR Ca Content

Increased CaMKII-dependent SR Ca leak is a major mechanism of CaMKII-dependent arrhythmias, and contractile dysfunction [35,36,43] is associated with the development of HF and can be detected in patients with AF [8,9,44,45,46,47]. CaMKII has been shown to phosphorylate the SR Ca release channel RyR2, resulting in increased diastolic open probability [48]. As a result, the frequency of spontaneous diastolic RyR2 openings (i.e., Ca sparks) is elevated, leading to increased diastolic SR Ca leak [49]. It is known that diastolic Ca release events activate the subsarcolemmal Na/Ca exchanger to export Ca out of the cell in exchange for Na [8]. This net transport of cations into the cell (3Na^+^ in exchange for 1 Ca^2+^) may result in delayed afterdepolarizations of the cell, which could serve as triggers for atrial arrhythmias [8]. We show here that the atria of patients with SDB undergo similar pathophysiological alterations. Enhanced SDB-dependent CaMKII activity results in increased SDB-dependent SR Ca leak, as measured by the frequency and characteristics of diastolic Ca sparks (Figure 3 and Appendix A). This is in accordance with increased CaMKII-dependent RyR2 phosphorylation in patients with SDB as previously shown [17]. Interestingly, in the present study the enhancement of SR Ca leak could be blocked by specific inhibition of CaMKII. 

Continuous diastolic SR Ca leak may result in reduced SR Ca content. As a consequence, less Ca is available to be released during systole, resulting in impaired contractile function [49]. Indeed, we report here a decreased SR Ca content in atrial myocytes of patients with SDB, which could be increased again by CaMKII inhibition (Figure 4). This is in contrast to atrial myocytes isolated from patients with persistent atrial fibrillation, which showed no difference in SR Ca content [8,9], but resembles more closely the situation for failing ventricular myocytes [49]. The underlying mechanism may be SDB-specific and involve dysfunction of SR Ca reuptake despite a possible CaMKII-dependent increase in phospholamban phosphorylation. The latter may just not be sufficient to overcome the increased SR-Ca leak, a similar situation as observed in heart failure [49].

Nevertheless, the SDB-dependent remodeling of atrial Ca handling would greatly increase the propensity for atrial arrhythmias like atrial fibrillation. Moreover, the reduced systolic SR Ca release due to reduced SR Ca content could impair atrial contractility and contribute to the formation of blood clots. Reduced blood flow velocities, particularly in the left atrial appendage, have been shown to be associated with cardioembolic stroke. Accordingly, we found in the present study that the prevalence of AF and stroke was significantly increased in patients with SDB (Table 1), consistent with previous data from community samples [11,50] and patients with chronic HF [51]. Interestingly, the prevalence of AF in the present SDB cohort was similar to the prevalence observed in patients with chronic HF [51].

### 4.3. SDB-Dependent Pro-Arrhythmic Signaling Is Independent of Pre-Existing AF or HF

As discussed above, CaMKII-dependent SR Ca leak has been observed in patients with AF [8,9]. Since atrial fibrillation alone might trigger pathophysiologic mechanisms independent of sleep apnea that would result in activation of CaMKII-dependent SR Ca leak, we aimed at excluding pre-existing AF as a potential confounder. First, multivariate linear regression indicated that the SDB-dependent increase in CaSpF was independent of pre-existing AF episodes, which is also consistent with previous findings [17]. Nevertheless, we have also studied patients in sinus rhythm separately in the present study. Interestingly, the SDB-dependent increase in CaMKII oxidation, activity and SR Ca leak was completely comparable to the entire patient cohort (Appendix A), indicating that pathophysiological factors triggered by SDB alone may be sufficient to oxidize and activate CaMKII, resulting in enhanced SR Ca leak.

In contrast to the clear association of SDB with CaMKII-dependent SR Ca leak, we found no significant correlation between magnitude of CaSpF and AF in the present study (Table 3). At first glance, this result appears to be counter-intuitive, since enhanced CaMKII-dependent SR Ca leak should also trigger episodes of AF. However, there are several explanations for this discrepancy. Firstly, we have not prospectively conducted Holter monitoring to identify patients with short lasting episodes of AF that depend on disturbed atrial Ca handling. The analysis of the prevalence of AF was solely based on retrospective evaluation of clinical information available at the time of surgery. Thus, paroxysmal AF may have been under-diagnosed in our patient cohort. Secondly, the development of AF is a complex process, since it involves early changes in Ca handling that predispose to atrial arrhythmias and is also accompanied by structural atrial remodeling, i.e., atrial fibrosis [52]. This aligns with our finding that the proportion of patients with a dilated left atrium was significantly higher in the SDB group (Table 1). Structural atrial remodeling may result in a form of AF, which may be independent of pro-arrhythmic atrial Ca handling. Therefore, the small sample size, and the potential inclusion of patients with structural atrial remodeling, may have obscured any clear association between AF and CaSpF in our study. Nevertheless, our findings clearly point to the value of more intensive monitoring of SDB patients for episodes of AF, especially considering the concomitant risk of debilitating AF complications like stroke.

In addition to pre-existing AF, systolic contractile dysfunction of the left ventricle may result in atrial structural and electrical remodeling independent of SDB. In human HF, diastolic SR Ca leak has been shown to be increased in the left ventricle and to contribute to contractile dysfunction and ventricular arrhythmias [36]. Because SDB is an independent risk factor for the development of HF [53] and, in our cohort, LVEF was significantly impaired in patients with SDB (Table 1), it seems important to exclude systolic contractile dysfunction as a confounding factor. However, in the present study, SDB-related diastolic SR Ca leak was independent of impaired EF (EF < 50%) in univariate and multivariate regression models (Table 3), consistent with previous data showing that atrial SDB-dependent RyR2 phosphorylation was independent of HF [17]. This indicates that atrial alterations directly mediated by SDB have a stronger effect than a possible influence of impaired ventricular function.

## 5. Conclusions

SDB is highly prevalent in patients undergoing CABG surgery. It is associated with a dilated left atrium, a history of AF and a history of stroke. We show here that both atrial oxidation and activation of CaMKII are increased in patients with SDB, resulting in an increased CaMKII-dependent SR Ca leak, independent of clinical confounders. Because the CaMKII-dependent SR Ca leak may increase the risk of AF in patients with SDB, it may represent a potential pharmaceutical target for the prevention or treatment of AF, especially since clinical CaMKII inhibitors are currently under development.

## Figures and Tables

**Figure 1 antioxidants-11-00331-f001:**
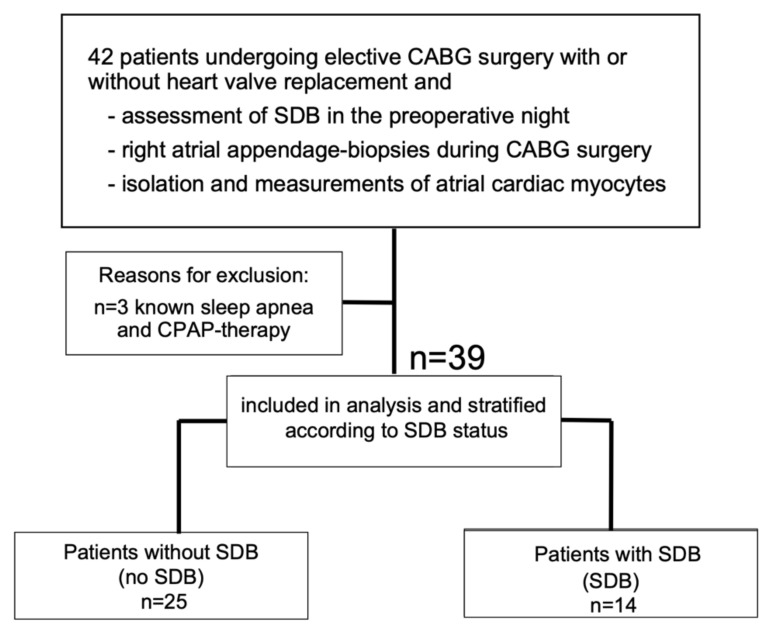
Patient flow chart. After screening, 39 patients were included in the analysis. Abbreviations: CABG, coronary artery bypass graft; CPAP, continuous positive airway pressure; SDB, sleep-disordered breathing.

**Figure 2 antioxidants-11-00331-f002:**
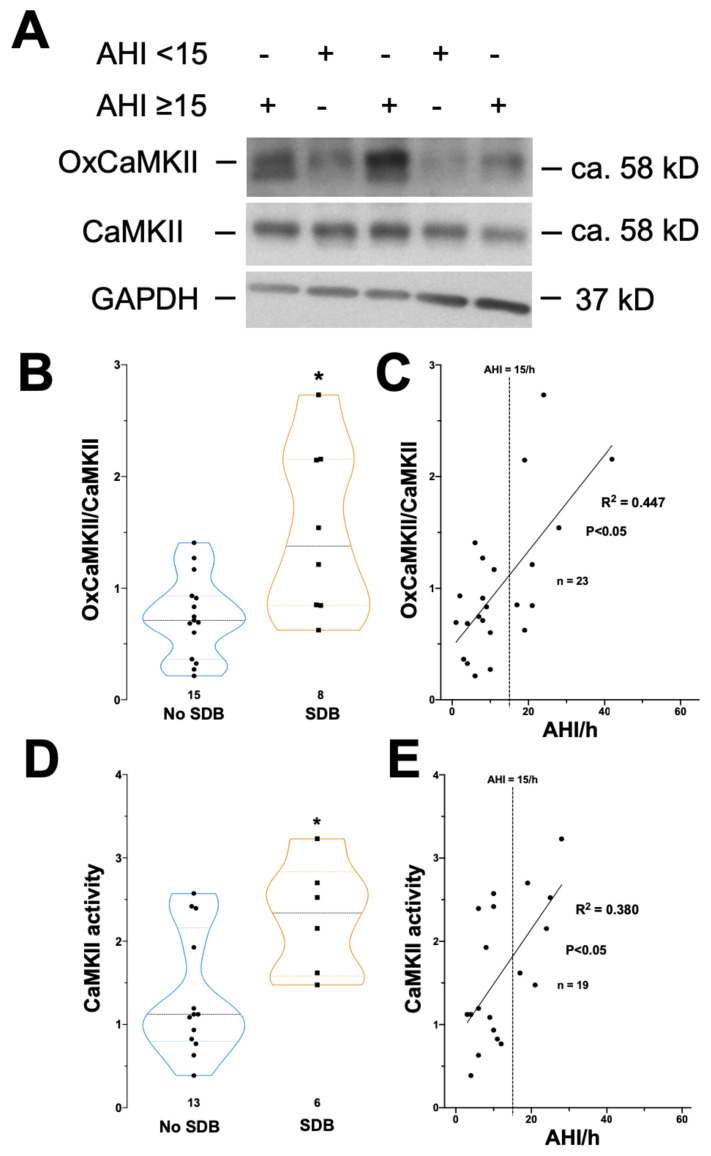
Increased oxidation and activity of CaMKII in patients with SDB. Original Western blot (**A**) and mean densitometric data (**B**) investigating the expression of oxidized CaMKII relative to CaMKII expression. Compared to control patients (no SDB), there was a significant increase in the level of oxidized CaMKII (oxCaMKII/CaMKII) in patients with SDB. (**C**) Scatter plot for the correlation of AHI and oxCaMKII/CaMKII. Linear regression analysis is shown as line plot. Mean densitometric data (**D**) for the assessment of CaMKII activity by using specific HDAC4 pulldown. There was a significant increase in CaMKII activity in patients with SDB. (**E**) Scatter plot for the correlation of AHI and CaMKII-activity. Linear regression analysis is shown as line plot. * *p* < 0.05 vs. no SDB (*t*-test).

**Figure 3 antioxidants-11-00331-f003:**
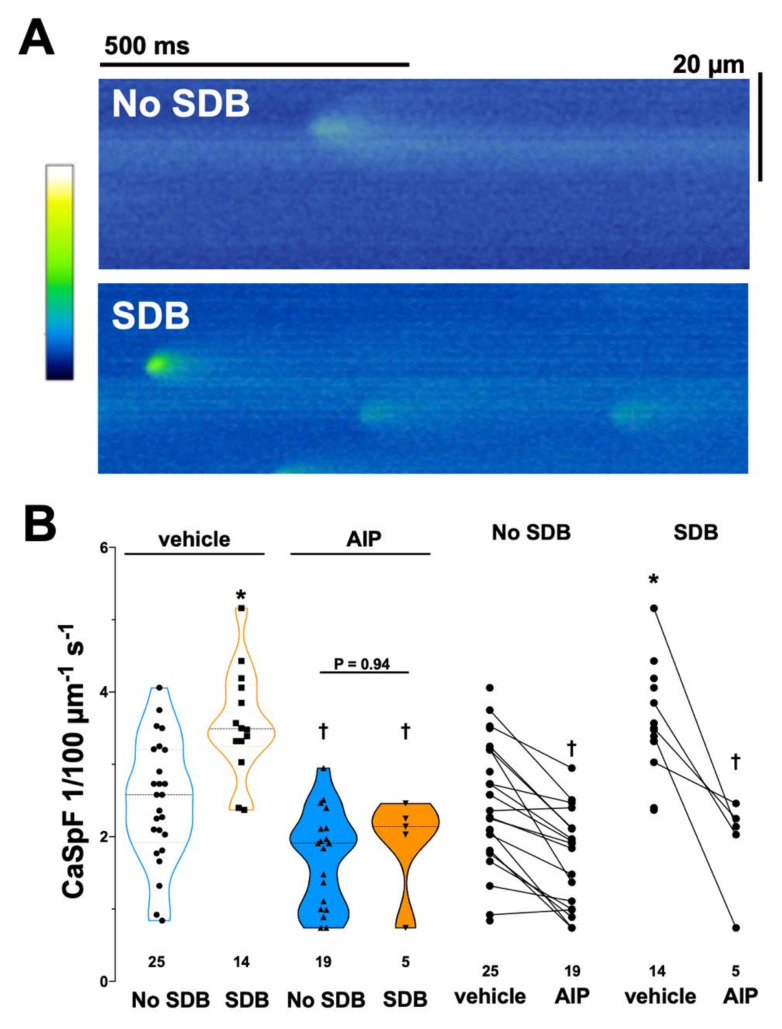
Patients with SDB showed increased Ca spark frequency. (**A**) Original confocal line scans of Fluo–4 loaded atrial cardiomyocytes isolated from patients without SDB and with SDB. The measured fluorescence intensity was color coded according to the calibration bar. (**B**) Mean data for diastolic Ca spark frequency (CaSpF) is shown as violin plot (left panel) or spaghetti plot (right panel). Compared to control patients (no SDB), the frequency of Ca sparks was significantly enhanced in patients with SDB. Importantly, this increase in CaSpF could be blocked with the selective CaMKII inhibitor autocamtide–2 related autoinhibitory peptide (AIP). * *p* < 0.05 vs. no SDB, † *p* < 0.05 vs. vehicle (two–way RM ANOVA, mixed effects model).

**Figure 4 antioxidants-11-00331-f004:**
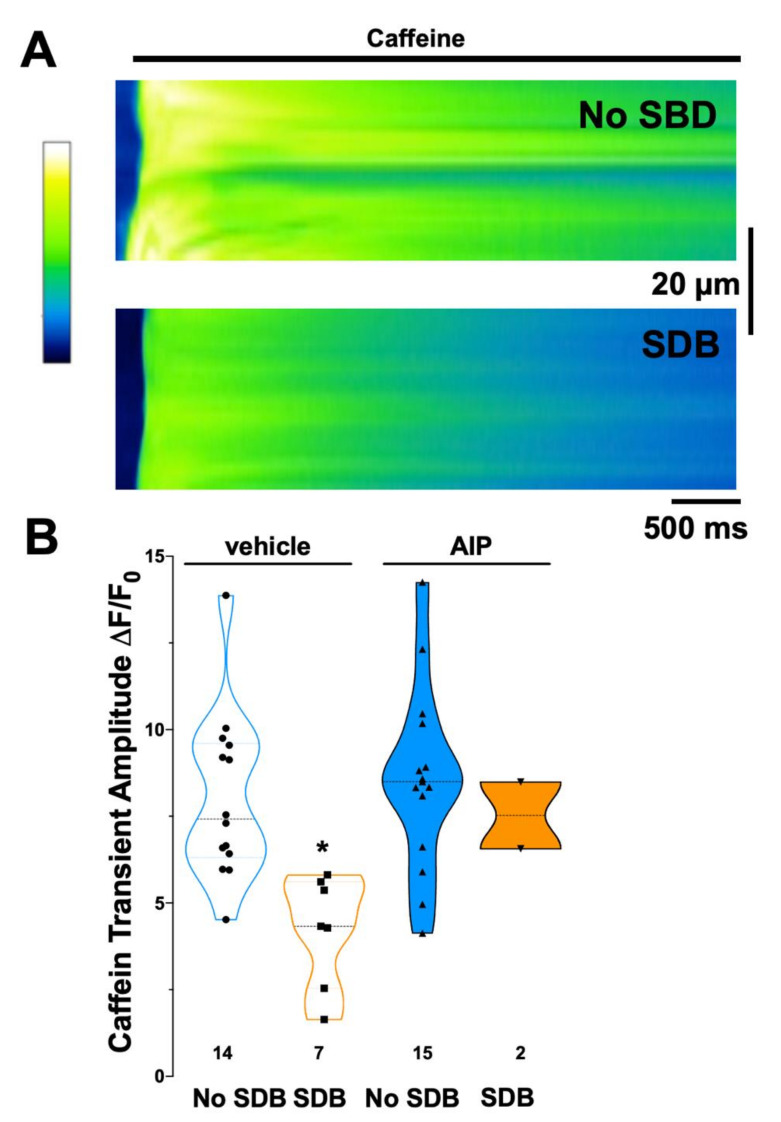
Patients with SDB display reduced SR Ca content. (**A**) Original confocal line scans of Fluo–4 loaded atrial cardiomyocytes rapidly exposed to caffeine (10 mmol/L). The measured fluorescence intensity was color coded according to the calibration bar. (**B**) Mean data for caffeine-induced Ca transient amplitude ΔF/F0 as a measure of SR Ca content. Compared to control patients (no SDB), the caffeine-transient amplitude was significantly lower in patients with SDB. Interestingly, the presence of AIP did not alter the caffeine-transient amplitude in patients without SDB. In contrast, in patients with SDB, there was a trend towards an increase in caffeine–transient amplitude in the presence of AIP. * *p* < 0.05 vs. no SDB (two–way RM ANOVA, mixed effects model).

**Table 1 antioxidants-11-00331-t001:** Baseline characteristics.

	No SDB (*n* = 25)	SDB (*n* = 14)	*p* Value
Age (years), mean ± SD	67 ± 10	70 ± 9	0.386
Male gender, n (%)	20 (80)	14 (100)	0.073
Body mass index (kg/m^2^), mean ± SD	28.8 ± 4.5	28.9 ± 4.5	0.962
CABG and valve replacement, n (%)	5 (20)	5 (36)	0.281
Cardiovascular risk factors			
Hypertension, n (%)	21 (84)	12 (86)	0.887
Diabetes mellitus, n (%)	6 (25)	5 (36)	0.482
Previous Smoker, n (%) *	10 (67)	6 (75)	0.679
Hypercholesterolemia, n (%)	16 (64)	8 (57)	0.673
Atrial fibrillation, n (%)	3 (12)	6 (43)	0.028
Paroxysmal AF	1 (4)	3 (21)	0.085
Persistent AF	2 (8)	3 (21)	0.229
Previous stroke n (%)	3 (12)	6 (42.9)	0.028
Heart and renal function			
Heart Failure, n (%)	15 (60)	10 (71)	0.475
NT-pro BNP (pg/mL), median (IQR)	226 (85, 1128)	1024 (414, 4575)	0.129
LVEF (%), mean ± SD **	55 ± 13	37 ± 11	0.004
GFR (mL/min), mean ± SD	77 ± 16	69 ± 28	0.262
Dilated left atrium, n (%) ***	3 (33)	9 (90)	0.011
Medication at admission			
ACE-inhibitors and/or AT1 blockers, n (%)	22 (88)	9 (64)	0.079
Betablockers	19 (76)	10 (71)	0.750
Statins, n (%)	20 (80)	8 (57)	0.128
Digitalis, n (%)	0 (0)	1 (7)	0.176
Aldosterone receptor antagonists, n (%)	3 (12)	2 (14)	0.838
Thiazid diuretics, n (%)	6 (24)	7 (50)	0.098
Loop diuretics, n (%)	6 (24)	6 (43)	0.221

IQR indicates interquartile range, which lies between the 25th and the 75th percentiles; sleep-disordered breathing is defined as an apnea–hypopnea index ≥ 15/h. CABG, coronary artery bypass grafting; NT-pro BNP, N-terminal pro-brain natriuretic peptide; LVEF, left ventricular ejection fraction; GFR, glomerular filtration rate; ACE, angiotensin-converting enzyme; AT1, angiotensin receptor. Divergent patient numbers from total n = 39: * current and previous smoker, n = 23 (n (AHI < 15) = 15; n (AHI ≥ 15 = 8); ** LVEF, n = 20 (n (AHI < 15) = 18; n (AHI ≥ 15 = 11); *** dilated left atrium, n = 19 (n (AHI < 15) = 9; n (AHI ≥ 15 = 10).

**Table 2 antioxidants-11-00331-t002:** Nocturnal polygraphy.

	No SDB (*n* = 25)	SDB (*n* = 14)	*p* Value
Total recording time, min	548 ± 91	516 ± 117	0.336
Apnea–hypopnea index, /h	6 ± 3	26 ± 13	<0.001
Apnea index, /h	2 ± 2	12 ± 13	0.012
Obstructive apnea index, /h	1 ± 1	5 ± 4	<0.001
Oxygen desaturation index (4%), /h	6.1 ± 3.3	24.1 ± 11.2	<0.001
Minimal oxygen saturation, %	79.1 ± 8.7	76.0 ± 5.9	0.249
Oxygen saturation below 90%, % of TRT	16 ± 20	24 ± 20	0.220
Cheyne–Stokes respiration, %/TRT	1 ± 3	9 ± 15	0.020

Data are mean ± SD. Sleep-disordered breathing (SDB) is defined as an apnea–hypopnea index ≥ 15/h. Abbreviations; TRT: total recording time.

**Table 3 antioxidants-11-00331-t003:** Linear regression analysis for CaSpF.

	Simple Linear Regression Analysis		Multiple Linear Regression Analysis			
			Model I	R^2^ 0.505	Model II	R^2^ 0.585
Variable	B (95% CI)	*p* Value	B (95% CI)	*p* Value	B (95% CI)	*p* Value
AHI, /h	0.052 (0.032; 0.071)	**<0.001**	0.048 (0.028; 0.068)	**<0.001**	0.040 (0.017; 0.062)	**<0.001**
ODI, /h	0.059 (0.038; 0.080)	**<0.001**				
MinO_2_, %	−0.021 (−0.061; 0.019)	0.288				
O_2_ below 90%, % of TRT	0.009 (−0.007; 0.024)	0.255				
Age/10, years	0.04 (−0.291; 0.370)	0.810	−0.002 (−0.257; 0.253)	0.985	0.025 (−0.255; 0.306)	0.855
Male gender	1.031 (0.153; 1.909)	**0.023**	0.591 (−0.140; 1.322)	0.110	0.703 (−0.049; 1.456)	0.066
Body-mass index, kg/m^2^	0.029 (−0.042; 0.100)	0.417	0.038 (−0.17; 0.093)	0.170	0.057 (−0.001; 0.115)	0.053
NT-pro BNP/1000, pg/mL	0.047 (−0.019; 0.114)	0.160			0.034 (−0.018; 0.087)	0.193
Diabetes	0.188 (−0.554; 0.790)	0.724			0.161 (−0.367; 0.690)	0.538
AF	0.169 (−0.550; 0.889)	0.636			−0.262 (−0.873; 0.349)	0.388
ACEi/ARB therapy	−0.799 (−1.532; −0.065)	**0.034**			−0.591 (−1.276; 0.094)	0.088
Beta blocker therapy	−0.081 (−0.802; 0.640)	0.821				
Valve replacement	−0.036 (−0.757; 0.686)	0.921				
LVEF < 50%	0.362 (−0.437; 1.160)	0.361				
Dilated left atrium	0.345 (0.726; 1.417)	0.506				

Model I accounts for age, male gender and BMI. Model II accounts for age, male gender, BMI, NT-pro BNP, diabetes, AF and ACEi/ARB therapy. Abbreviations: AHI: apnea–hypopnea index, ODI: oxygen desaturation index, MinO_2_: minimal oxygen saturation, O_2_ below 90%: oxygen saturation below 90%, TRT: total recording time, NT-pro BNP: N-terminal pro-brain natriuretic peptide, AF: atrial fibrillation, LVEF: left ventricular ejection fraction, ACEi/ARB: ACE inhibitor/angiotensin-receptor blocker therapy.

**Table 4 antioxidants-11-00331-t004:** Linear regression analyses for caffeine-transient amplitude (measure of SR Ca load).

	**Simple Linear Regression** **Analysis**		**Multiple Linear Regression Analysis**			
			**Model I**	**R^2^ 0.343**	**Model II**	**R^2^ 0.436**
**Variable**	**B (95% CI)**	***p* Value**	**B (95% CI)**	***p* Value**	**B (95% CI)**	***p* Value**
AHI, /h	−1.042 (−1.794; −0.291)	**0.009**	−1.048 (−1.952; −0.144)	**0.026**	−0.123 (−0.241; −0.005)	**0.043**
ODI, /h	−0.133 (−0.213; −0.053)	**0.003**				
MinO_2_, %	0.075 (−0.061; 0.211)	0.264				
O_2_ below 90%, % of TRT	−0.045 (−0.097; 0.008)	0.090				
Age/10, years	−0.252 (−1.723; 1.219)	0.724	−0.588 (−2.088; 0.911)	0.418	0.317 (−2.284; 2.918)	0.795
Male gender	−1.915 (−4.764; 0.934)	0.176	−0.503 (−3.568; 2.562)	0.732	0.585 (−4.114; 5.285)	0.791
Body-mass index, kg/m^2^	−0.010 (−0.306; 0.286)	0.945	0.054 (−0.255; 0.364)	0.715	0.112 (−0.296; 0.519)	0.562
NT-pro BNP/1000, pg/mL	−0.141 (−0.352; 0.071)	0.181			−0.112 (−0.406; 0.183)	0.426
Diabetes	0.201 (−2.507; 2.908)	0.878			0.808 (−2.305; 3.921)	0.582
AF	−0.883 (−3.662; 1.896)	0.515			−1.716 (−6.904; 3.474)	0.485
ACEi/ARB therapy	0.254 (−2.430; 2.937)	0.846			−1.313 (−4.922; 2.297)	0.444
Beta blocker therapy	−2.071 (−4.577; 0.436)	0.100				
Valve replacement	−2.125 (−4.754; 0.503)	0.107				
LVEF < 50%	−1.585 (−3.654; 0.483)	0.123				
Dilated left atrium	−2.342 (−5.508; 0.824)	0.130				

Model I accounts for age, male gender and BMI. Model II accounts for age, male gender, BMI, NT-pro BNP, diabetes, AF and ACEi/ARB therapy. Abbreviations: AHI: apnea–hypopnea index, ODI: oxygen desaturation index, MinO_2_: minimal oxygen saturation, O_2_ below 90%: oxygen saturation below 90%, TRT: total recording time, NT-pro BNP: N-terminal pro-brain natriuretic peptide, AF: atrial fibrillation, LVEF: left ventricular ejection fraction, ACEi/ARB: ACE inhibitor/angiotensin-receptor blocker therapy.

## Data Availability

The authors declare that all method protocols used in this study are available for any researcher upon request. To exclude the possibility of unintentionally sharing private patient information, patient data can only be made available after informed consent about a specific request has been given by each patient.

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
