# Peer review of "Enhanced Cardiac CaMKII Oxidation and CaMKII-Dependent SR Ca Leak in Patients with Sleep-Disordered Breathing"

_antioxidants, 2022, doi:10.3390/antiox11020331_

Round 1
Reviewer 1 Report
This is a follow up paper from the work performed by the same group published in Circ Res over a year ago, where patients waiting for CABG surgery were monitored the night before surgery for SBD. In their seminal work, the authors noticed increased CaMKII oxidation, increased RyR Ca leak (although not as thoroughly or directly studied as in the current manuscript) and increased late Na current all that could be acutely attenuated by CaMKII inhibition. In this study, the authors have confirmed that patients with SBD waiting for CABG surgery show a propensity for increased RyR Ca leak in human atrial tissue samples that can be acutely attenuated by CaMKII inhibition. The paper for the most part is well-written and a quick read. The novelty of the findings is somewhat dampened by the earlier publication. There are also some concerns with the quality of the westerns. Below are some suggestions/comments that might be useful to improve the manuscript.
- There does not appear to be any difference in the lowest O2 saturation or the percent of time below 90%. The chosen patient population although stratified by SBD seem to be overall similarly hypoxic? Some discussion and explanation seem warranted for these findings.
- Although there appears to be a statistical increase in spark frequency in the SBD group (even less so in calculated leak), this increase does not appear to be as large in magnitude as the observed decrease in SR load. How might the authors explain the apparent mismatch in SR spark frequency/leak to the loss in SR Ca load? The authors speculated a decrease in SERCA activity but wouldn’t the increased CAMKII activity enhance PLB phosphorylation too?
- What other proteins, besides CaMKII, may also be excessively oxidized? It seems one way to better differentiate the current manuscript from the previous manuscript would be to dig a bit deeper into other protein modifications in addition to CaMKII?
- The representative loading control (GAPDH) in Fig 2 appears to unusually and substantially increase across the gel. It is unclear if the samples are variably loaded or if there was an issue with the detection of GAPDH for this particular western. The total CaMKII western does not appear to follow the same pattern. There is concern that the westerns are not as carefully performed as need to be for the quantifications performed in the manuscript.
- There are a lot of abbreviations in the abstract, some not defined (ie. AHI) that make the abstract challenging to read. It would be helpful to find a way to reduce these abbreviations for the general reader.
- It is not clear why the authors sometimes chose a simple T-test vs multi-linear regression analysis to compare groups?
- Are SBD and AHI the same group?
Reviewer 2 Report
This study suggests that oxidative stress causes CaMKâ…¡-dependent SR Ca leak, which contributes to the pathogenesis of atrial fibrillation in patients with sleep-disordered breathing. The research is done properly and the results contain new findings and are significant.
Major comment
In Table1, No SDB patients tend to have a higher proportion of ACE inhibitor/ARB medication. It is possible that the reduction of preload by these may affect left atrial enlargement and LVEF. Have you analyzed the association between ACE inhibitor/ARB and CaMKâ…¡oxidation and CaSpF ?
Minor
What is the data of the following sentences from line 298 “Accordingly, we found that the prevalence of AF and stroke was significantly increased in patients with SDB, consistent with previous data …” ?
In line 109, authors state that NT-proBNP levels tended to be higher in the SDB group, but the p-value is 0.129 in Table 1. is an overstatement.
In line 42, references 7-9 are more than 7 years old, not recent.
It is better to use the same notation for scale bars, etc. in Figures 3A and 4A.
